# The 2016 California policy to eliminate nonmedical vaccine exemptions and changes in vaccine coverage: An empirical policy analysis

**Sindiso Nyathi**[1‡], **Hannah C. Karpel**[2‡], **Kristin L. Sainani**[1], **Yvonne Maldonado**[1,3], **Peter J. Hotez**[4,5,6,7], **Eran Bendavid**[8], **Nathan C. Lo**[9]*

1 Department of Epidemiology and Population Health, Stanford University School of Medicine, Stanford, California, United States of America, 2 New York University School of Medicine, New York, New York, United States of America, 3 Division of Infectious Diseases, Department of Pediatrics, Stanford University School of Medicine, Stanford, California, United States of America, 4 Texas Children's Hospital Center for Vaccine Development, Departments of Pediatrics and Molecular Virology and Microbiology, Baylor College of Medicine, Houston, Texas, United States of America, 5 Department of Biology, Baylor University, Waco, Texas, United States of America, 6 Hagler Institute for Advanced Study at Texas A&M University, College Station, Texas, United States of America, 7 James A. Baker III Institute for Public Policy, Rice University, Houston, Texas, United States of America, 8 Center for Population Health Sciences, Division of Primary Care and Population Health, Department of Medicine, Stanford University, Stanford, California, United States of America, 9 Department of Medicine, University of California, San Francisco, San Francisco, California, United States of America

‡ These authors are co-first authors on this work.
* Nathan.Lo@ucsf.edu

## Abstract

### Background

Vaccine hesitancy, the reluctance or refusal to receive vaccination, is a growing public health problem in the United States and globally. State policies that eliminate nonmedical ("personal belief") exemptions to childhood vaccination requirements are controversial, and their effectiveness to improve vaccination coverage remains unclear given limited rigorous policy analysis. In 2016, a California policy (Senate Bill 277) eliminated nonmedical exemptions from school entry requirements. The objective of this study was to estimate the association between California's 2016 policy and changes in vaccine coverage.

### Methods and findings

We used a quasi-experimental state-level synthetic control analysis and a county-level difference-in-differences analysis to estimate the impact of the 2016 California policy on vaccination coverage and prevalence of exemptions to vaccine requirements (nonmedical and medical). We used publicly available state-level data from the US Centers for Disease Control and Prevention on coverage of measles, mumps, and rubella (MMR) vaccination, nonmedical exemption, and medical exemption in children entering kindergarten. We used county-level data individually requested from state departments of public health on overall vaccine coverage and exemptions. Based on data availability, we included state-level data

**Data Availability Statement:** The data and code files underlying the results of the study are

available from the Github code repository (https://github.com/SindisoNyathi/California-Vaccine-Policy) and Figshare data repository (DOI: https://doi.org/10.6084/m9.figshare.9775496.v1).

**Funding:** The authors received no specific funding for this work. NCL was previously supported by the Medical Scientist Training Program (Stanford University School of Medicine).

**Competing interests:** I have read the journal's policy and the authors of this manuscript have the following competing interests: NCL reports funding from the World Health Organization for work outside of the current study. No other authors have competing interests.

**Abbreviations:** ACS, American Community Survey; CDC, Centers for Disease Control and Prevention; CI, confidence interval; DTaP, diphtheria, tetanus, and pertussis; MMR, measles, mumps, and rubella; SB277, Senate Bill 277; STROBE, Strengthening the Reporting of Observational Studies in Epidemiology.

for 45 states, including California, from 2011 to 2017 and county-level data for 17 states from 2010 to 2017. The prespecified primary study outcome was MMR vaccination in the state analysis and overall vaccine coverage in the county analysis.

In the state-level synthetic control analysis, MMR coverage in California increased by 3.3% relative to its synthetic control in the postpolicy period (top 2 of 43 states evaluated in the placebo tests, top 5%), nonmedical exemptions decreased by 2.4% (top 2 of 43 states evaluated in the placebo tests, top 5%), and medical exemptions increased by 0.4% (top 1 of 44 states evaluated in the placebo tests, top 2%). In the county-level analysis, overall vaccination coverage increased by 4.3% (95% confidence interval [CI] 2.9%–5.8%, $p < 0.001$), nonmedical exemptions decreased by 3.9% (95% CI 2.4%–5.4%, $p < 0.001$), and medical exemptions increased by 2.4% (95% CI 2.0%–2.9%, $p < 0.001$). Changes in vaccination coverage across counties after the policy implementation from 2015 to 2017 ranged from −6% to 26%, with larger increases in coverage in counties with lower prepolicy vaccine coverage. Results were robust to alternative model specifications. The limitations of the study were the exclusion of a subset of US states from the analysis and the use of only 2 years of postpolicy data based on data availability.

## Conclusions

In this study, implementation of the California policy that eliminated nonmedical childhood vaccine exemptions was associated with an estimated increase in vaccination coverage and a reduction in nonmedical exemptions at state and county levels. The observed increase in medical exemptions was offset by the larger reduction in nonmedical exemptions. The largest increases in vaccine coverage were observed in the most "high-risk" counties, meaning those with the lowest prepolicy vaccine coverage. Our findings suggest that government policies removing nonmedical exemptions can be effective at increasing vaccination coverage.

## Author summary

### Why was this study done?

- Vaccine hesitancy, often parental reluctance or refusal to vaccinate their children, is a growing challenge in public health; however, there is debate on the role of government-level vaccination policies to address this problem.

- Limited empirical research has evaluated the impact of governmental vaccination policies that restrict the ability of parents to obtain nonmedical exemptions from routine vaccination requirements.

- The 2016 California vaccine policy (Senate Bill 277) that eliminated nonmedical childhood vaccination exemptions provides an opportunity to empirically estimate the impact of a governmental state-level policy on vaccine coverage.

**What did the researchers do and find?**

- We evaluated the 2016 California vaccine policy by applying a synthetic control analysis with state-level data and a difference-in-differences analysis with county-level data to estimate the relationship between the policy and changes in measles, mumps, and rubella (MMR) or overall vaccine coverage, nonmedical exemptions, and medical exemptions.

- At the state level, the California vaccine policy was associated with a 3.3% increase in MMR vaccination coverage, a 2.4% decrease in nonmedical exemptions, and a 0.4% increase in medical exemptions.

- At the county level, the California vaccine policy was associated with a 4.3% increase in overall vaccination, a 3.9% decrease in nonmedical exemptions, and a 2.4% increase in medical exemptions. The largest increases in postpolicy county-level coverage occurred in counties with lower prepolicy vaccine coverage.

**What do these findings mean?**

- Our study found that the 2016 California policy to eliminate nonmedical childhood vaccination exemptions was associated with an increase in vaccination coverage and a decrease in nonmedical exemptions.

- The study findings support the hypothesis that government policies can be an effective tool to increase vaccination coverage, especially in the most "high-risk" (low vaccine coverage) settings.

## Introduction

Vaccine hesitancy, defined as the reluctance or refusal to vaccinate despite the availability of vaccinations, is a growing public health challenge in the US and globally [1–3]. The decline in vaccination rates, driven by vaccine hesitancy and lack of confidence in vaccines, has led to recent outbreaks of vaccine-preventable diseases and threatens the public health gains made against these infectious diseases over past decades [4–6]. The factors driving vaccine hesitancy are complex and include misconceptions and misinformation about vaccine safety, low perceived risk of infectious diseases, and lack of trust in healthcare providers [7, 8].

The policy debate surrounding vaccine hesitancy in the US has focused on vaccine exemptions, which provide an option for parents to waive current vaccination requirements for entry into school or daycare [9–11]. Currently, 18 states allow nonmedical exemptions to school vaccine requirements based on philosophical, personal, or other beliefs [4, 12]. Notably, all states permit medical exemptions to school immunization requirements for clinical conditions in which vaccination would be contraindicated (e.g., immunosuppression and receipt of live attenuated vaccines). Recent work indicates that in states allowing these nonmedical exemptions, the numbers of nonmedical exemptions are as high as 15%–25% in some counties. In 2019, measles emerged in seven of the 14 urban counties previously identified as high risk due to nonmedical exemptions [7]. However, there continues to be debate surrounding the

effectiveness of policies that restrict nonmedical exemptions, with ongoing legislative discourse across the US [12].

Following a series of high-profile exposures and outbreaks of vaccine-preventable diseases—including a measles outbreak in Orange County, CA—California passed Senate Bill 277 (SB277), which eliminated a previous policy that allowed nonmedical exemptions to school immunization requirements [13–17]. Although previous observational studies have suggested that coverage increased in California after the policy's implementation, the policy's effectiveness remains unclear given natural variation in vaccination rates and lack of controlled policy evaluation [18–21]. Furthermore, the rate of medical exemptions in California increased after the policy, causing concerns that children who had received nonmedical exemptions were instead receiving medical exemptions, thus limiting the policy's ability to increase overall vaccine coverage [7, 22].

The aim of our study was to conduct a rigorous, controlled analysis to estimate the effectiveness of California's policy restricting nonmedical exemptions to increase vaccination coverage. To address ongoing debate surrounding these governmental policies, we provide a quasi-experimental, controlled analysis to evaluate the association between California's policy and vaccination and exemption outcomes.

## Methods

### Data sources and study outcomes

We performed two empirical analyses to estimate the association of the 2016 California policy with changes in vaccination coverage and exemptions: (1) a synthetic control analysis, in which we used state-level data to create a hypothetical counterfactual "synthetic control California" to compare with the real California; and (2) a difference-in-differences analysis, in which we compared county-level data from California and control states. The California policy (SB277) went into effect on July 1, 2016, corresponding with the 2016–2017 school year. Our study objectives, methods, and planned analyses were prespecified in a preanalysis plan (S3 Appendix) [23].

For the state-level synthetic control analysis, we used publicly available state-level data from the US Centers for Disease Control and Prevention (CDC) SchoolVaxView website [24]. These data are collected by state immunization programs and aggregated by the CDC [25, 26]. Data were available from 2009 to 2017 (S1 Appendix). Data for the 2010–2011 school year were unavailable because they were not verified, so only data from 2011 onward were used in this analysis. The primary outcome for the state-level analysis was prespecified as measles, mumps, and rubella (MMR) vaccine coverage, defined as the proportion of children entering kindergarten who have received the two-dose MMR vaccine by the start of the school year. The secondary outcomes were the proportion of children entering kindergarten at the start of the school year with medical exemptions or the proportion entering kindergarten with nonmedical exemptions. We included medical and nonmedical exemptions as outcomes to evaluate whether any compensatory increases in medical exemptions were observed to be associated with the policy. We used a range of publicly available state covariate data (S1 Table). We obtained demographic data (e.g., education, income) from the US Census Bureau [27]. Data on health-related characteristics were retrieved from the Data Resource Center for Child and Adolescent Health and the Centers for Medicare and Medicaid Services [28, 29]. All results are presented as absolute percentage changes in MMR coverage and medical and nonmedical exemptions.

For the county-level difference-in-differences analysis, we used county-level data individually requested from state departments of public health. These data are compiled by county

health departments using data submitted by public and private schools and then submitted to state departments of public health. We contacted health departments in all 50 states and the District of Columbia to request data. We received full county-level data from California and 16 other states for overall vaccination coverage and 17 states for vaccination exemptions (S4 Fig). When counties did not report overall vaccine coverage, we used county-level MMR coverage as a proxy for overall coverage. Missing county-level outcomes data were either imputed or excluded from the analysis, depending on the proportion of missing data (S2 Appendix). The primary outcome for the county-level analysis was prespecified as overall vaccination coverage, defined as the proportion of children entering kindergarten with all required vaccines at the start of the school year. The immunizations required to enroll in kindergarten vary by state but generally include MMR, polio, chicken pox (varicella), and diphtheria, tetanus, and pertussis (DTaP) [30]. The secondary outcomes were the proportion of children entering kindergarten at the start of the school year with medical exemptions or the proportion entering kindergarten with nonmedical exemptions. We used covariate data from the American Community Survey (ACS) [31]. In our final dataset, we only included counties with a population of at least 65,000 based on availability of covariate data. Additional information on covariate selection is described in S2 Appendix. All results are presented as absolute percentage changes in overall vaccination coverage and medical and nonmedical exemptions.

The primary outcome was MMR vaccine coverage in the state-level analysis and overall vaccine coverage in the county-level analysis. This difference in primary outcomes between analyses was prespecified and based on data availability because states do not report overall vaccine coverage to the CDC. We assumed state-level MMR coverage was a good proxy for overall coverage and stated this in our preanalysis plan (S3 Appendix). Furthermore, because the definitions of medical and nonmedical exemptions include children with exemptions for any vaccine and not just the MMR vaccine, for any given state the three outcomes did not always sum to one. In both the state- and county-level analyses, we used data for children entering kindergarten. Kindergarten is the school grade for children ages 5–6 years in the US and commonly the entry level into school. This age group is most likely to be affected by the policy change given the age of the children in relation to immunization requirements. Although the 2016 California policy also requires younger-aged children in preschool or daycare to have completed all the relevant vaccinations, kindergarten is the beginning of formal education and, as such, local and state health departments routinely collect immunization data for children in kindergarten.

## Statistical analysis

We used a synthetic control study design to estimate the relationship between the California policy and vaccination coverage and exemptions at the state level. The synthetic control method is a statistical tool designed for comparative case studies, such as policy evaluations, in which only a single treated unit is available (e.g., state-level policy) [32–34]. The approach constructs a hypothetical control state (i.e., a "synthetic control California") that matches the treated state (i.e., actual California) on the prepolicy outcome. The resulting synthetic control California provides a counterfactual estimation of the study outcome during the postpolicy period in the treated state in the absence of the treatment—i.e., it projects the outcome for California in the absence of the vaccine policy. The synthetic California is constructed with a weighted combination of control states. The synthetic control optimization algorithm estimates a weight for all nonexperimental states that minimizes the difference between the actual California and the synthetic California in the pretreatment period. Most states receive a zero

weight, and only states with nonzero weights provide information for the construction of the synthetic control.

We created a synthetic control for California using untreated states as potential controls. The synthetic California was created by matching on average prepolicy outcomes (i.e., vaccine coverage and prevalence of exemptions) and demographic and health-related characteristics. These characteristic covariates were chosen using a stepwise variable selection process and cross-validation procedure to avoid overfitting (S1 Appendix). States were excluded from the construction of the synthetic control if they were missing relevant data. The key effect size for each study outcome was the difference in pre- to postpolicy change between California (i.e., the treated state) and the synthetic control California (i.e., the hypothetical untreated state). We constructed a unique synthetic control for each of the three outcomes, per convention given the data-driven nature of the process. This resulted in three synthetic control California states—one corresponding to each of the three outcomes (one primary, two secondary).

To assess whether the effect size in the synthetic control analysis was meaningful, we used conventional placebo testing. Whereas regression models provide confidence intervals (CIs) and *p*-values based on frequentist assumptions, inference in synthetic control methods is grounded in placebo tests (also known as permutation tests). In placebo tests, we reevaluate the effect size under the null condition (i.e., repeating the synthetic control method for each of the untreated states). The resulting placebo effect sizes for each of the control states quantify the variation in the outcome under the null hypothesis. By comparing them to the actual (California) effect size, we can determine whether the observed effect size in the treated unit is meaningful or whether it is similar in magnitude to the variation in the outcome in the absence of a treatment. The latter case would imply that our intervention or policy is not statistically meaningful [35, 36]. To conduct the placebo tests, we individually reassigned treatment status to each of the other states in the control pool. We then created a synthetic control for the new state and assessed the resulting effect size for every state in the dataset. We compared effect sizes for each state relative to California for all outcomes (S1 Appendix). We prespecified an effect size as meaningful if California was in the top fifth percentile of states.

We used a difference-in-differences study design to evaluate the association between the California policy and vaccination coverage and exemptions at the county level. The difference-in-differences design estimates the relative change in vaccination coverage and exemptions over time associated with the California policy as the difference between the treated group of counties (California) before and after the policy implementation and the control counties (counties not in California) before and after the policy implementation. To assess the parallel trends assumption required for the difference-in-differences methodology, we plotted the data for California and the control counties before the policy's implementation (S2 Appendix). We used an ordinary least squares regression model for all outcomes and calculated robust standard errors clustered by county. We included an adjusted analysis with prespecified county-level characteristics (S2 Appendix).

## Sensitivity analysis

We conducted sensitivity analyses to evaluate the robustness of our models (S1 Appendix and S2 Appendix) [37]. In the synthetic control and difference-in-differences analyses, we reran the models, iteratively excluding a single state from the control pool. These "leave-one-out" tests allowed us to evaluate the influence of individual states in the control pool on the effect size. For the county-level analysis, we performed a subanalysis in which we analyzed the data from states reporting complete overall vaccination coverage and excluding states that reported only MMR coverage. Finally, for the state-level analysis, we reran the model with varying

combinations of characteristic covariates to evaluate the influence of the covariate combination on the outcome.

This study is reported per the Strengthening the Reporting of Observational Studies in Epidemiology (STROBE) guidelines (S4 Appendix). All statistical analyses were conducted in R (version 3.5.1). Public datasets and analytical materials are available online (S1 Table) [38, 39]. This was not human subjects research as the study relied on publicly available, secondary and aggregated data sources and was exempt from Stanford University's Institutional Review Board approval.

## Results

We included state-level vaccination and exemption data for 45 states, including California, from 2011 to 2017 (states excluded due to missing data are shown in S2 Table). The average prepolicy MMR vaccination coverage at the state level in the 2015 school year was 94.2% and ranged across states from 87.1% to 99.4% (94.5% in California). The average prepolicy nonmedical exemption prevalence in 2015 was 2.2% and ranged across states from 0.4% to 6.2% (2.4% in California); the average prepolicy medical exemption prevalence was 0.3% and ranged across states from 0.1% to 1.2% (0.2% in California).

We included county-level vaccination and exemption data for 17 states from 2010 to 2017 based on availability of data from state departments of public health (S6 Table). In California, the mean prepolicy county-level vaccination coverage in 2015 was 92.6% (95% CI 91.8–93.4) and ranged across counties from 68.9% to 100%, the prepolicy county average for nonmedical exemption prevalence was 3.1 (95% CI 2.5–3.6) and ranged across counties from 0% to 21.9%, and the prepolicy county average for medical exemption prevalence was 0.21 (95% CI 0.18–0.23) and ranged across counties from 0% to 1.7%.

### State-level analysis

For each of the three state-level outcomes, the synthetic control California matched the observed California in the prepolicy period (Fig 1). The synthetic control California for each of the three outcomes was predominantly composed of three to five control states, with all other states having minimal weights (Fig 2). No single state had a weight greater than 50% except for Texas in the synthetic California for the nonmedical exemptions outcome. In this state-level analysis, we estimated that the 2016 California policy was associated with a 3.3% increase in MMR coverage, from 94.5% in 2015, relative to its synthetic control in the postpolicy period (top 2 of 43 states evaluated in the placebo tests, top 5%). The largest improvement in MMR coverage occurred in North Dakota (3.6%). The 2016 California policy was associated with a 2.4% decrease in nonmedical exemptions in the postpolicy period (top 2 of 43 states evaluated in the placebo tests, top 5%), from 2.4% in 2015, whereas medical exemptions increased by 0.4% (top 1 of 44 states evaluated in the placebo tests, top 2%), from 0.2% in 2015 (Fig 3). The largest decrease in nonmedical exemptions occurred in Vermont (2.8%). California had the largest increase in medical exemptions, followed by Maryland (0.2% increase).

### County-level analysis

In the county-level analysis, we found that the trends for outcome variables were relatively similar between California counties and counties in control states in the prepolicy period (S5 Fig). Overall vaccination was generally similar in the prepolicy period, although California counties had slightly lower vaccination levels and greater variation than control counties. Prepolicy variation in nonmedical exemptions for California was greater than for control states (S7 Table). In the county-level analysis, we estimated that the California policy was associated

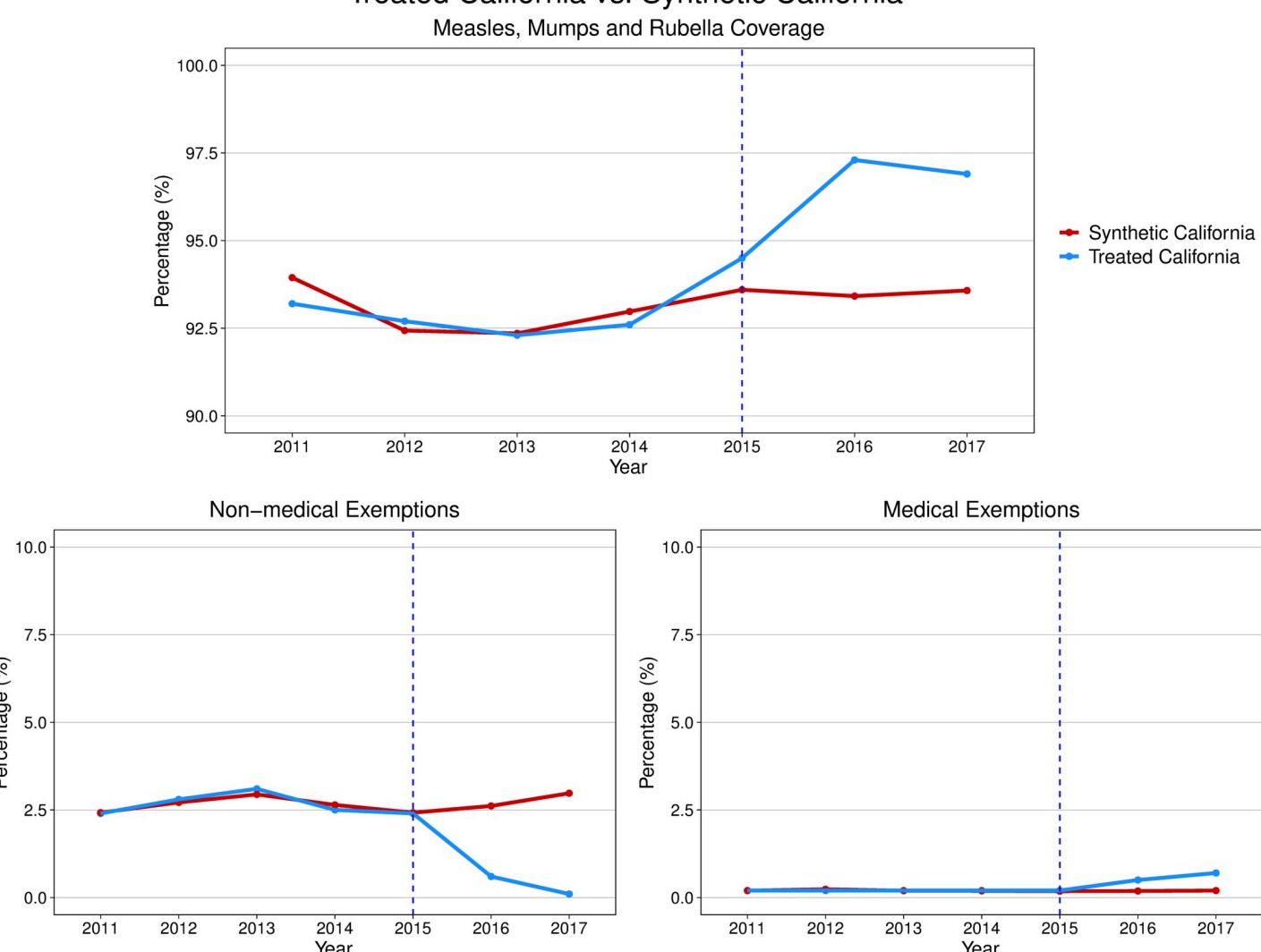

**Fig 1. State-level analysis of the 2016 California policy using synthetic control methodology on childhood vaccination and exemption outcomes.** We constructed a synthetic control state for California for three study outcomes of MMR coverage (top), nonmedical exemptions (bottom left), and medical exemptions (bottom right). The synthetic control state (red) is estimated to match the study outcomes for actual California (blue) before the policy (indicated by dotted blue line, which is before the 2016–2017 school year). The figure shows the observed difference between the outcome in the actual treated California and the synthetic control California after the policy. MMR, measles, mumps, and rubella.

with a 4.3% (95% CI 2.9–5.8, $p < 0.001$) absolute increase in vaccine coverage for children entering kindergarten in California compared with those in control states (Table 1). The policy was also associated with a 3.9% (95% CI 2.4–5.4, $p < 0.001$) absolute decrease in nonmedical exemptions and a 2.4% (95% CI 2.0–2.9, $p < 0.001$) absolute increase in medical exemptions compared with counties in control states (Table 1). We also estimated coefficients for the characteristic demographic covariates included in the model. Income and education status of less than a high school degree were associated with increases in vaccination coverage and decreases in medical and nonmedical exemptions (Table 1). The percentage of uninsured children was significantly associated with a small increase in medical exemptions. Education status of some college as well as bachelor's degree or beyond were significantly associated with small decreases in nonmedical exemptions, whereas race (defined as percent white, based on prior literature

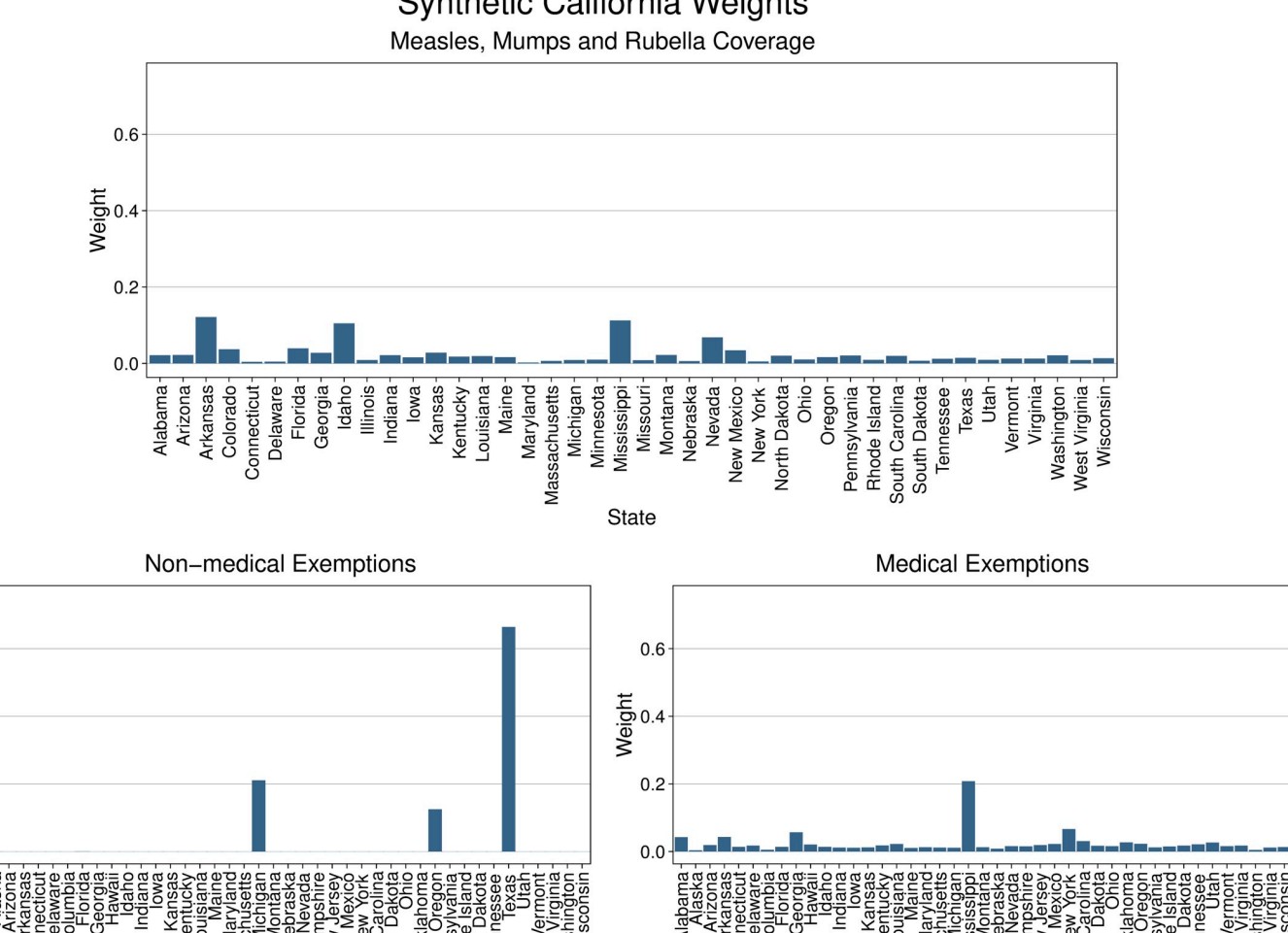

**Fig 2. Weights of the control states to construct the "synthetic control California" for state-level analysis of three study outcomes.** The synthetic control California was constructed as a hypothetical control state from a weighted sum of the untreated states (i.e. the control pool). The weights were calibrated to match the prepolicy vaccine outcome for California. This figure shows the weights of the control states for each of the three outcomes: MMR coverage (top), nonmedical exemptions (bottom left), and medical exemptions (bottom right). MMR, measles, mumps, and rubella.

[40]) and percent of uninsured children were significantly associated with small increases in nonmedical exemptions (Table 1).

There was variation in vaccination coverage and exemptions across counties before and after the policy implementation. Counties with a higher proportion of prepolicy nonmedical exemptions in 2015 (i.e., "hot spots" most at risk of outbreaks) had larger decreases in non-medical exemptions following the policy's implementation (Fig 4). Likewise, counties with lower prepolicy overall coverage had the largest increases in overall coverage following the policy's implementation. The absolute change in vaccination coverage between 2015 and 2017 across counties in California ranged from −6% to 26%, with 12/57 (21.1%) of counties decreasing coverage, 28/57 (49.1%) experiencing an increase in coverage between 0.1 and 4 percentage points, and 17/57 (29.8%) experiencing an increase in coverage >4 percentage points (S8

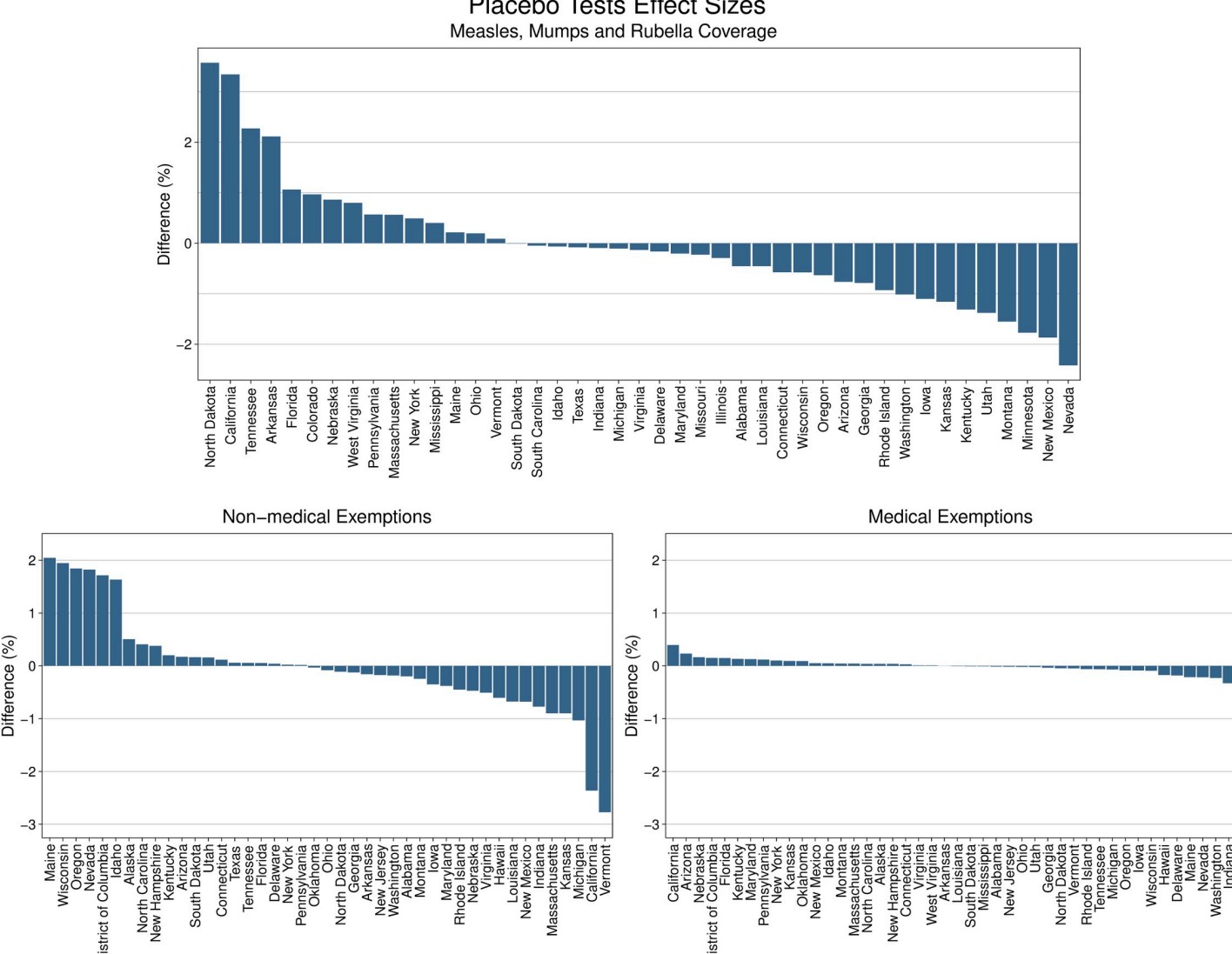

**Fig 3. Placebo testing for the state-level analysis of the 2016 California policy using synthetic control methodology.** Placebo testing is used in synthetic control analyses to determine whether an identified relationship is statistically meaningful. We prespecified a meaningful relationship for the 2016 California policy as a change in the hypothesized direction (increase in vaccination coverage, decrease in nonmedical exemption, increase in medical exemption) in the top fifth percentile of all states. Placebo testing was done by repeating the methodology for each of the control states individually to estimate the percent change in vaccine coverage or exemptions that would be measured if the policy occurred in the control state the same year as California; we then compared the estimated effect sizes. The estimated percent change by state is presented for three study outcomes of MMR coverage (top), nonmedical exemptions (bottom left), and medical exemptions (bottom right). MMR, measles, mumps, and rubella.

Table). For nonmedical exemptions, 10/57 (17.5%) of counties in California either had no change or an increase, and 47/57 (82.5%) of counties had a decrease in nonmedical exemptions ranging from −0.1% to −21.5% between 2015 and 2017. All counties had an absolute increase in medical exemptions, ranging from 0.5% to 9.3%. Plumas County, population 18,742, experienced the 9.3% rise in medical exemptions, from 1.0% in 2015 to 10.3% in 2017 (S8 Table).

## Sensitivity analysis

The study findings were overall robust to various sensitivity analyses. In both analyses, changing the control data included did not change the overall findings (S5 Table and S9 Table). In

**Table 1. County-level analysis of changes in vaccination coverage associated with the 2016 California policy using a difference-in-differences regression.**

| Parameter | Overall vaccination coverage (95% CI)[a] | p-Value | Nonmedical exemptions prevalence (95% CI)[b] | p-Value | Medical exemptions prevalence (95% CI)[b] | p-Value |
|---|---|---|---|---|---|---|
| **2016 California policy[b]** | 4.3 (2.9–5.8) | <0.001 | −3.9 (5.4–2.4) | <0.001 | 2.4 (2.0–2.9) | <0.001 |
| **Median income per US$10,000 (no)** | 1.0 (0.9–2.0) | <0.001 | −0.3(0.4–0.08) | 0.005 | −0.3 (0.4–0.1) | 0.002 |
| **Mean household size (no)** | −0.1 (−2.3 to 2.1) | 0.9 | 0.2 (−0.6 to 1.0) | 0.7 | 0.5 (−0.3 to 1.3) | 0.3 |
| **Population per 100,000 (no)** | −0.1 (−0.9 to 0.6) | 0.7 | 0.01 (−0.03 to 0.04) | 0.7 | 0.03 (−0.02 to 0.08) | 0.2 |
| **Poverty per 1,000 (%)** | −0.4 (−4.6 to 3.8) | 0.8 | −0.5 (−2.0 to 1.0) | 0.5 | 0.05 (−0.9 to 1.0) | 0.9 |
| **White (%)** | −0.03 (−0.06 to 0.01) | 0.2 | 0.03 (0.01–0.04) | 0.001 | 0.003 (−0.01 to 0.01) | 0.6 |
| **Education: Less than high school (%)** | 0.5 (0.2–0.7) | <0.001 | −0.3 (0.4–0.2) | <0.001 | −0.09 (0.1–0.04) | <0.001 |
| **Education: Some college or less (%)** | 0.2 (−0.03 to 0.4) | 0.1 | −0.2 (0.2–0.09) | <0.001 | −0.02 (−0.07 to 0.02) | 0.3 |
| **Education: Bachelor's degree or higher (%)** | 0.07 (−0.2 to 0.3) | 0.7 | −0.1 (0.2–0.09) | <0.001 | −0.01 (−0.06 to 0.05) | 0.8 |
| **Uninsured children (%)** | 0.08 (−0.01 to 0.2) | 0.1 | 0.09 (0.04–0.1) | <0.001 | 0.06 (0.04–0.08) | <0.001 |

[a]Robust standard errors, clustered by county.

[b]Difference-in-differences estimates represent relative change in county-level vaccination and exemption prevalence for children entering kindergarten in California before and after the 2016 policy compared with children entering kindergarten before and after the policy in counties from control states.

Abbreviations: CI, confidence interval; no, number

the state-level analysis, the effect size was robust to changes in the characteristic covariates used in the model (S4 Table). In the county-level analysis, the subanalysis excluding states reporting MMR coverage did not alter the effect size (S10 Table). Details on the sensitivity analysis are included in S1 Appendix and S2 Appendix.

## Discussion

In this empirical policy analysis, we evaluated California's 2016 policy removing nonmedical vaccine exemptions and found evidence that the policy was associated with an increase in vaccination coverage in children entering kindergarten. At the state level, the policy was associated with a 3.3% increase in MMR coverage and a 2.4% reduction in nonmedical exemptions. The policy was also associated with a 0.4% increase in medical exemptions. At the county level, often the counties with high baseline exemption prevalence and most "at risk" of an outbreak had the largest increases in vaccine coverage from the policy. Despite concerns around the observed increase in medical exemptions, our study found that the policy was associated with an overall increase in vaccine coverage. The use of two independent state- and county-level analyses with consistent results provides evidence to support considering the adoption of similar governmental policies eliminating nonmedical exemptions to help address the growing public health challenge of vaccine hesitancy across the United States and globally.

Previous descriptive analyses examined the impact of the California policy shortly after its implementation. These analyses found an increase in overall vaccination coverage among students from 92.8% in 2015 to 95.1% in 2017 along with a decrease in nonmedical exemptions [7, 22]. However, the policy's effectiveness remained unclear because these analyses did not account for variation and trends in vaccine coverage across the US. There was also an increase in medical exemptions (from 0.2% in 2015–2016 to 0.7% in 2017–2018) that might have offset the described decreases in nonmedical exemptions [18, 22]. Two recent studies provided additional descriptive analyses of the impact of the California vaccine policies on vaccination

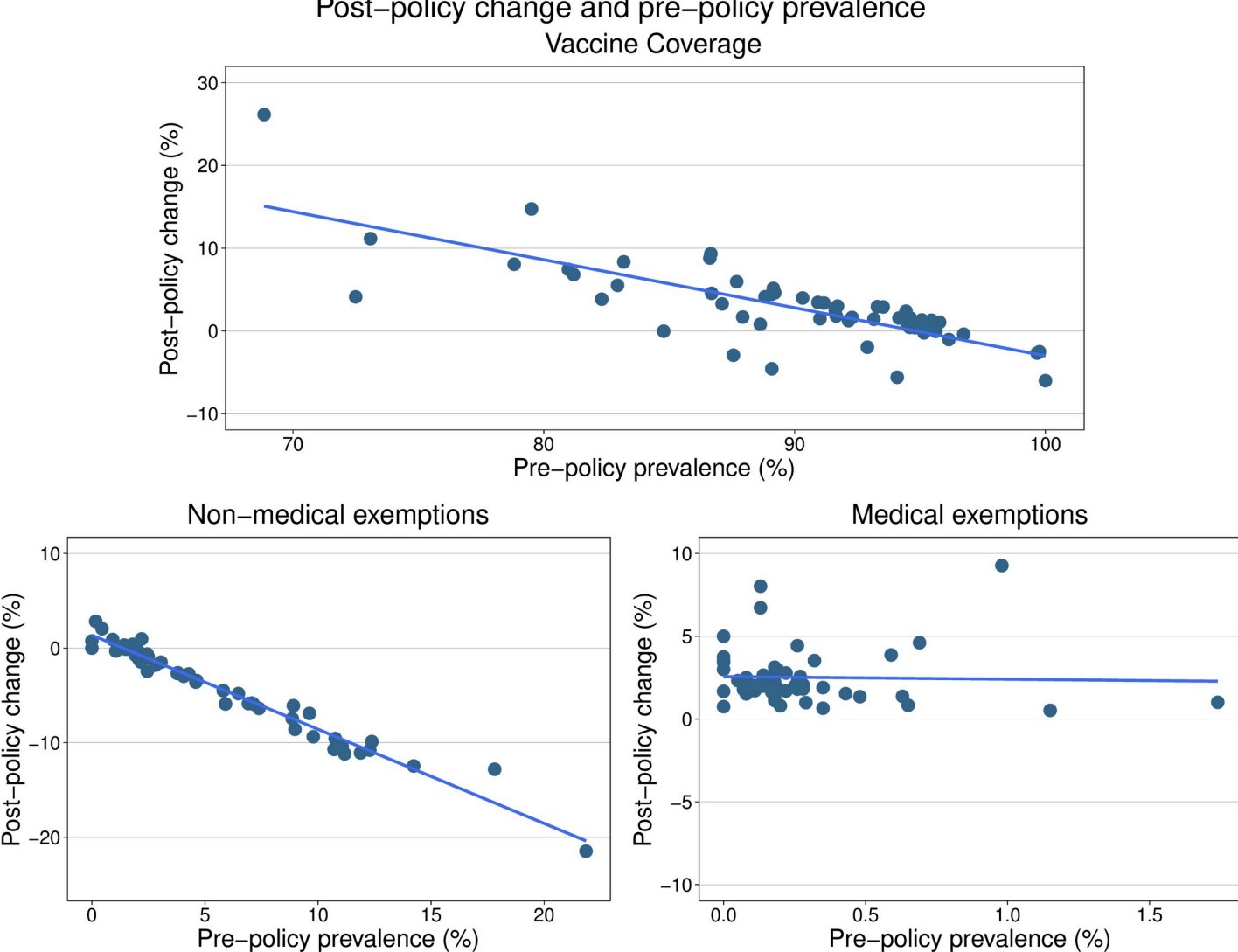

**Fig 4. Postpolicy changes in county-level vaccination coverage and exemptions as a function of prepolicy prevalence.** This figure plots the change in overall vaccination coverage and exemptions before and after the policy for each county in California as it related to their prepolicy (2015) prevalence. In counties with greater prepolicy prevalence of nonmedical exemptions, there were larger decreases in nonmedical exemptions following the policy. Counties with lower prepolicy overall vaccination coverage had greater changes in overall coverage following the policy's implementation. Note that the axis magnitudes are different for each plot based on baseline magnitude of outcome.

coverage [20, 21]. In one study, the authors used county-level data for California to report the changes in the percentage of children entering kindergarten with up-to-date vaccination status, as well as geographic clustering of unvaccinated children associated with three California vaccination policies. In a subsequent study, the authors extended this work to focus specifically on SB277, the 2016 California vaccine policy. Both studies similarly found an increased percentage of children entering kindergarten with all vaccinations completed postimplementation of the California policy, though attribution to SB277 is limited by availability of comparison units to form a control group. This leaves open the possibility that observed changes may be over- or underestimated, depending on the secular trends. Our study further extended this work by providing a controlled, quasi-experimental methodological approach to estimate the impact of the 2016 California vaccine policy. Taken together, these studies contribute robust

evidence to support the conclusion that vaccination coverage increased after the California policy's implementation.

In this study, in both the county- and state-level analyses, nonmedical exemptions decreased and medical exemptions increased, a trend that supports our conclusion that the observed increase in vaccination coverage in California is associated with the implementation of the 2016 policy rather than confounding factors. Additionally, although the rise in medical exemptions could indicate that some children who may have received nonmedical exemptions in the past are now receiving medical exemptions, the net effect following the California policy was still an increase in vaccination coverage. Notably, new legislation was recently approved in California to require centralization of medical exemptions, similar to policies in place in Mississippi and West Virginia—states that also do not allow nonmedical exemptions for school entry. A centralized review may further increase vaccination coverage in California [12].

In our state-level synthetic control analysis, we used placebo testing to evaluate whether the estimated changes in study outcomes (e.g., vaccine coverage) associated with the California policy were statistically meaningful. The observed increase in MMR coverage in California associated with the 2016 policy was much greater than the changes in the majority of the placebo states and met our prespecified threshold for a significant finding (Fig 3, S3 Fig). However, we also observed notable changes in a select number of placebo states. North Dakota had an increase in coverage of 3.6% relative to its synthetic control during the year of the policy— 0.3% larger than the increase in California attributable to the policy. We did not find evidence that North Dakota implemented any statewide vaccination policies at this time. The observed increase could be due to the state's continuing efforts to address vaccine hesitancy through other mechanisms. Vermont had the largest decrease in nonmedical exemptions, 2.8% relative to its synthetic control during the year of the policy—0.4% larger than the decrease seen in California. This is likely due to Vermont's 2015 policy, which removed philosophical exemptions [41]. Notably, the number of children enrolled in kindergarten in North Dakota and Vermont in 2016 was approximately 10,000 and 6,500, respectively, whereas the kindergarten enrollment for California in the same year was approximately 521,000 [31]. Therefore, despite the modestly larger percentage changes in smaller states like North Dakota and Vermont, the absolute number of children affected is much larger in California. Nonmedical exemptions in California showed a slight decrease in 2014. This may have been a result of another California policy passed in 2014 that required parents seeking exemptions to provide signed documentation from a healthcare provider [42].

Although the state-level increase in vaccination coverage was modest (approximately 3%), this estimate should be interpreted in the context of county-level variation in vaccine coverage and risk of outbreaks. Mainly, California counties with higher prevalence of prepolicy nonmedical exemptions and most at risk for outbreaks had larger improvements in overall coverage. Existing evidence suggests that the potential for infectious disease outbreaks is driven by hot spots of low vaccine coverage, meaning some counties may have particularly low coverage and be prone to outbreaks [4, 7, 12]. States with laws permitting personal belief exemptions or easily obtainable exemptions have greater hot spots, with high levels of nonmedical exemptions and outbreak risk, compared to states with stricter exemption laws [12]. When examining county-level data, we found a range of effect sizes for the change in vaccination coverage between 2015 and 2017, from a 6% decline in Sierra County to a 26% increase in Trinity County (S8 Table). The county-level variation in the absolute change in overall vaccination coverage and exemptions suggests that the effectiveness of the policy may be higher in most at-risk (i.e., low coverage) hot spots. The variation in the change in vaccination coverage across counties before and after the policy implementation may help explain why the increase in overall coverage does not equal the difference between the reduction in nonmedical

exemptions and the increase in medical exemptions. This discrepancy may also be because our definitions of the outcome variables as the state- or county-reported vaccination and exemption percentages do not reflect missing certificate or voucher data from schools.

The importance of the observed increases in vaccine coverage in California after the implementation of the policy is further illustrated by considering how small changes in coverage have disproportionate effects on the number and size of outbreaks, as well as on public health costs [6]. A recent modeling study predicted that a 5% decline in MMR vaccine coverage driven by vaccine hesitancy would result in a 3-fold increase in measles cases in children per year, corresponding with a cost of US $2.1 million [4]. Additionally, the benefit of small increases in vaccine coverage is particularly evident when approaching approximated herd immunity levels of 90%–95% vaccine coverage. In 2015, 24 California counties had coverage levels below the range needed for herd immunity; in 2017, this number was 12 counties. This finding has implications for other states, as current data suggest that many states have counties and communities below the conventional threshold for herd immunity [4]. As multiple states continue to experience outbreaks of vaccine-preventable diseases, interventions like the California policy that increase vaccine coverage levels above the required thresholds will continue to be vital. A recent analysis of the California policy suggested that legislation may need to consider local differences because areas of northern California continued to demonstrate high rates of students without up-to-date vaccination status despite the state's intervention [20]. Our county-level analysis found a relationship between income and education, and vaccine coverage, suggesting that socio-demographic factors may play an important role in vaccine coverage. Additional research into other correlates of vaccine hesitancy in particular regions may be necessary to obtain ideal coverage in areas that remain vaccine hesitant. Finally, settings that remain at low vaccine coverage despite removal of nonmedical exemptions may require additional interventions that include local community involvement and educational programs [43–45].

The study results should be interpreted in the context of the limitations of the data and study design. Vaccine coverage and exemption data are reported by schools, and data collection procedures vary by state and year. We included a subset of US states in the analysis (included sample size of $N = 45$ for state data, $N = 16–17$ for county data) based on data availability, which could potentially introduce bias. To address these limitations, we used two different datasets and methodologies that both aim to control for differences between California and control states and counties in order to assess the effectiveness of the California policy [45]. We also ran a set of sensitivity analyses that showed that our findings were robust to the states included in the control pool. For both the state- and county-level analyses, the statistical design assumed that no concurrent change occurred in 2016. To address this, we conducted a brief literature review to determine whether any concurrent interventions took place during this time and did not find any potentially confounding programs [46, 47]. The synthetic control is ideally constructed without any acute changes during the preintervention period, although California passed two policies aiming to increase vaccine coverage. In 2014, California passed AB2109, which required proof of having consulted a healthcare provider before receiving an exemption. In 2015, California introduced a second initiative meant to educate administrators on the conditional admissions requirements. The 2015 conditional entrants initiative, although effective, was limited in scope compared with the statewide policy and would only affect MMR coverage or overall exemptions and not nonmedical exemptions. In addition, these policies were unlikely to meaningfully affect our estimate because the calibration process of the synthetic control minimized these perturbations [42]. We plotted prepolicy trends in vaccination coverage and exemptions to assess the parallel trends assumption in the difference-in-differences analysis. Although the trends were similar between California and control

counties before the intervention, there was greater variability in California over the years, especially for nonmedical exemptions. Finally, our analyses included only 2 years of postpolicy data, so we are unable to observe the persistence of increased coverage over time.

As vaccine hesitancy becomes a larger public health challenge in the US and globally, with vaccine-preventable disease outbreaks growing, the debate concerning state policies to remove nonmedical exemptions is ongoing. Although such policies do not address the larger problem of vaccine hesitancy, including a lack of confidence in vaccination, they effectively increase vaccination coverage. Our study finds that vaccine coverage increased and nonmedical exemptions decreased in the time period following the implementation of the California policy eliminating nonmedical exemptions. Our conclusions are strengthened by the use of two independent analyses, which both found similar results. These study results support the idea that state-level governmental policies to remove nonmedical exemptions can be effective strategies to increase vaccination coverage across the US.

## Supporting information

**S1 Appendix. The synthetic control method and state-level data.**
(DOCX)

**S2 Appendix. The difference-in-differences method and county-level data.**
(DOCX)

**S3 Appendix. Preanalysis plan.**
(DOCX)

**S4 Appendix. STROBE checklist.** STROBE, Strengthening the Reporting of Observational Studies in Epidemiology.
(DOCX)

**S1 Fig. Characteristic state covariate selection cutoffs.**
(DOCX)

**S2 Fig. Cross validation of synthetic controls using training and testing data for variable selection.**
(DOCX)

**S3 Fig. Placebo plot trajectories for control states (gray) and treated state (blue).**
(DOCX)

**S4 Fig. Flow chart of data collection for county-level outcome data from state health departments for control states.**
(DOCX)

**S5 Fig. Prepolicy trends in outcome variables in California and control counties.**
(DOCX)

**S1 Table. Data sources for state-level synthetic control analysis.**
(DOCX)

**S2 Table. States excluded from control pool due to missing data.**
(DOCX)

**S3 Table. Characteristic covariate weights used to create the synthetic California for each outcome in base case analysis.**
(DOCX)

**S4 Table. Characteristic covariate sensitivity analysis for state-level synthetic control analysis.**
(DOCX)

**S5 Table. Control state sensitivity analysis with leave-one-out tests.**
(DOCX)

**S6 Table. List of control states for the county-level difference-in-differences analysis for each outcome.**
(DOCX)

**S7 Table. Prepolicy trends for outcome variables in California and control states.**
(DOCX)

**S8 Table. Change in county-level outcome variables between 2015 and 2017 for California counties.**
(DOCX)

**S9 Table. County-level sensitivity analysis with leave-one-out tests.**
(DOCX)

**S10 Table. County-level sensitivity analysis with a subset of data reporting overall vaccine coverage.**
(DOCX)

## Acknowledgments

We would like to thank state departments of public health for providing us with county-level vaccine coverage and exemption data and for their ongoing inspirational work to improve vaccination coverage and public health.

## Author Contributions

**Conceptualization:** Nathan C. Lo.

**Data curation:** Sindiso Nyathi, Hannah C. Karpel, Nathan C. Lo.

**Formal analysis:** Sindiso Nyathi, Hannah C. Karpel, Nathan C. Lo.

**Investigation:** Sindiso Nyathi, Hannah C. Karpel, Nathan C. Lo.

**Methodology:** Sindiso Nyathi, Hannah C. Karpel, Eran Bendavid, Nathan C. Lo.

**Project administration:** Nathan C. Lo.

**Resources:** Kristin L. Sainani, Yvonne Maldonado, Peter J. Hotez, Eran Bendavid, Nathan C. Lo.

**Software:** Sindiso Nyathi, Hannah C. Karpel, Nathan C. Lo.

**Supervision:** Kristin L. Sainani, Yvonne Maldonado, Peter J. Hotez, Eran Bendavid, Nathan C. Lo.

**Validation:** Sindiso Nyathi, Hannah C. Karpel, Kristin L. Sainani, Yvonne Maldonado, Peter J. Hotez, Eran Bendavid, Nathan C. Lo.

**Visualization:** Sindiso Nyathi, Hannah C. Karpel, Nathan C. Lo.

**Writing – original draft:** Sindiso Nyathi, Hannah C. Karpel, Nathan C. Lo.

**Writing – review & editing:** Sindiso Nyathi, Hannah C. Karpel, Kristin L. Sainani, Yvonne Maldonado, Peter J. Hotez, Eran Bendavid, Nathan C. Lo.

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
