## [Decision Letter · Decision Letter 0]

29 Aug 2019

Dear Dr. Nyathi,

Thank you very much for submitting your manuscript "Effectiveness of the 2016 California Policy to Eliminate Non-Medical Exemptions on Vaccine Coverage" (PMEDICINE-D-19-02566) for consideration at PLOS Medicine. 

Your paper was evaluated by a senior editor and discussed among all the editors here. It was also discussed with an academic editor with relevant expertise, and sent to four independent reviewers, including a statistical reviewer. The reviews are appended at the bottom of this email and any accompanying reviewer attachments can be seen via the link below:

[LINK]

In light of these reviews, I am afraid that we will not be able to accept the manuscript for publication in the journal in its current form, but we would like to consider a revised version that addresses the reviewers' and editors' comments. Obviously we cannot make any decision about publication until we have seen the revised manuscript and your response, and we plan to seek re-review by one or more of the reviewers. 

We expect to receive your revised manuscript by Sep 19 2019 11:59PM. Please email us (plosmedicine@plos.org) if you have any questions or concerns.

We look forward to receiving your revised manuscript. 

Sincerely,

Caitlin Moyer, Ph.D.

Associate Editor 

PLOS Medicine

plosmedicine.org

1.Did your study have a prospective protocol or analysis plan? Please state this (either way) early in the Methods section.

c) In either case, changes in the analysis—including those made in response to peer review comments—should be identified as such in the Methods section of the paper, with rationale.

2. Thank you for agreeing to make your data available. The provided Github link (from Reference 29) to the dataset does not seem to work. At this time, please provide the corrected link to the data repository and accession numbers required for access.

3. Please revise your title according to PLOS Medicine's style. Your title must be nondeclarative and not a question. It should begin with main concept if possible. "Effect of" should be used only if causality can be inferred, i.e., for an RCT. Please place the study design ("A randomized controlled trial," "A retrospective study," "A modelling study," etc.) in the subtitle (ie, after a colon).

4. Abstract: Please quantify the main results with 95% CIs and p values.

5. Abstract: Please address the study implications without overreaching what can be concluded from the data; the phrase "In this study, we observed ..." may be useful. Specifically, please avoid causal language such as “The California policy resulted in…”. 

6. Abstract: Please avoid vague statements such as "these results have major implications for policy/clinical care". Mention only specific implications substantiated by the results. Specifically, the phrase “meaningful increase” in the Conclusion section is vague.

8. Results: Please provide the CIs and p values for the comparison of actual vs synthetic CA analysis and county-level analysis results for post-policy changes in vaccine coverage and medical/nonmedical exemptions. Please also provide p-values for Table 1.

9. Figure S1: Please define the abbreviation “RMSPE“. Also, please show the axis beginning at zero. If this is not possible, please show a break in the axis.

10. Figure S2: Please show the axis beginning at zero. If this is not possible, please show a break in the axis.

11. Table S7: Please specify if these values are percentages.

12. Tables S9 and S10: Please provide any p-values associated with 95% CIs.

13. Please ensure that the study is reported according to the relevant guidelines, which can be found here: http://www.equator-network.org/

Please include the completed STROBE, RECORD, etc. checklist as Supporting Information. When completing the checklist, please use section and paragraph numbers, rather than page numbers. Please add the following statement, or similar, to the Methods: "This study is reported as per the Strengthening the Reporting of Observational Studies in Epidemiology (STROBE) guideline (S1 Checklist)."

Comments from the reviewers:

Reviewer #1: See attachment

Michael Dewey

Reviewer #2: This is a well-conducted and well-reported study on the effect of the 2016 California law removing non-medical exemptions in that state. 

There are a few issues that need to be addressed to improve this manuscript.

1. In the abstract and in the introduction, you refer to vaccine hesitancy as the delay or refusal of vaccination, but hesitancy can also mean underlying lack of confidence in vaccination, even if vaccines are received. This may be a bit pedantic, but hesitance does not always equate to refusal or delay.

2. Throughout the manuscript, it should be more clear whether the % changes that are presented are relative or absolute changes. It appears they are absolute changes, but as written, it is a bit confusing.

3. A little more methodologic explanation of the synthetic control analysis would be helpful, especially for policy makers who may read this who do not have more advanced statistical training.

4. There are some reported results where % change is given without the context of baseline rates. While this is presented well in Figure 4 for the county-level analysis, it would be helpful to ensure this has the best context throughout.

5. In the discussion, third paragraph, it would be helpful to mention that ND and VT are relatively small states, compared to CA, and relatively small changes in absolute numbers of children vaccinated or exempted could make a larger impact than in CA. Additionally, some context as to the overall absolute change, in N, of the children with vaccination or without exemption, would be helpful.

Reviewer #4: This analysis evaluates the effectiveness of SB277 in California. There have been two recent publications tackling the exact same question (below). This paper does use different set of methods than those. 

Pingali SC, Delamater PL, Buttenheim AM, Salmon DA, Klein NP, Omer SB. Associations of Statewide Legislative and Administrative Interventions With Vaccination Status Among Kindergartners in California. JAMA. 2019;322:49-56.

Delamater PL, Pingali SC, Buttenheim AM, Salmon DA, Klein NP, Omer SB. Elimination of Nonmedical Immunization Exemptions in California and School-Entry Vaccine Status. Pediatrics. 2019;:e20183301.

Specific concerns/issues/comments

The synthetic control approach is very interesting, but I do have some reservations about its validity in this particular scenario. There are numerous factors that can and do influence state-level UTD rates, but specifically policy changes are shocks that may interrupt and/or change prior temporal trends. In California, prior to SB277, the state enacted AB2109 two years prior to SB277, which interrupted temporal trends in the year prior (note the small decreases in NMEs in 2014 and 2015 in Fig 1. It seems that constructing a synthetic control without accounting for the different policy regimes in the potential control data over the time period (i.e., other states) may lead to an inaccurate estimate of what we should have expected in CA.

The analysis does not consider the other avenues for not up to date students to enter school in CA (conditional, not subject to immunization requirements). And, unfortunately in this case, UTD rates is not an appropriate metric to evaluate the effectiveness of SB277 because efforts other than SB277 influenced UTD rate. Specifically, the analysis does not consider the statewide effort (that occurred one year prior to SB277) to reduce Conditional entrants, which resulted in a dramatic reduction of conditional entrants statewide and a corresponding increase in UTD rate (this can be seen in Fig 1 from 2014 to 2015). The Delamater et al. paper goes into great detail on this matter.

The explanatory portion of the analysis seems somewhat underdeveloped, as the results in Table 1 are not really given much attention.

[LINK]

---

## [Decision Letter · Decision Letter 1]

28 Oct 2019

Dear Dr. Nyathi,

Thank you very much for re-submitting your manuscript "Impact of the 2016 California Policy to Eliminate Non-Medical Exemptions on Vaccine Coverage: An Empirical Policy Analysis" (PMEDICINE-D-19-02566R1) for review by PLOS Medicine.

I have discussed the paper with my colleagues and the academic editor and it was also seen again by 2 reviewers. I am pleased to say that provided the remaining editorial and production issues are dealt with we are planning to accept the paper for publication in the journal.

[LINK]

We look forward to receiving the revised manuscript by Nov 04 2019 11:59PM. 

Sincerely,

Caitlin Moyer, Ph.D.

Associate Editor 

PLOS Medicine

plosmedicine.org

Requests from Editors:

1. Title: Thank you for revising your title to be consistent with PLOS Medicine’s style. To remove any confusion regarding potential causal association, please revise to “Estimated association of the 2016 California Policy to Eliminate Non-Medical Exemptions on Vaccine Coverage: An Empirical Policy Analysis” or similar.

2. Abstract: Methods and Findings: Please briefly mention the years for which data were included for the state level and county-level analyses, e.g. from your results: “We included state level vaccination and exemption data for 45 states, including California, from 2011 to 2017…” and “We included county level vaccination and exemption data for 17 states from 2010 to 2017 based on availability of data from state health departments…”.

3. Abstract: Methods and Findings: Please clarify briefly in the text what is meant by the presentation of results such as “(top 2 of 43 states, top 5%)”- does this indicate that post-policy implementation that the synthetic California ranks 2 out of 43 states included in the analysis?

4. Abstract: Conclusions: Please revise the conclusions to avoid causal and imprecise language, For the first sentence, we suggest: “The California policy was associated with an estimated increase in vaccination coverage and reduction in non-medical exemptions at state and county levels.” or similar. 

5. Abstract: Conclusions: Please revise the final sentence to “Our findings suggest that government policies removing non-medical exemptions may be effective at increasing vaccination coverage.” or “The results of this study suggest that government policies removing non-medical exemptions may be effective at increasing vaccination coverage.” or similar.

6. Author summary: “Why was the study done?”: Please revise the first point to “...a growing challenge in public health; however, there is debate surrounding the role of state level vaccination policies…” or similar, to avoid unclear language. Specifically, please remove the term “substantial” from the first bullet point, as the meaning is imprecise.

7. Author summary: “What did the researchers do and find?”: Please combine the first two points, e.g. “We evaluated the 2016 California vaccine policy that eliminated non-medical childhood vaccination exemptions by applying a synthetic control analysis using state level data and a difference-in-differences analysis using county level data to estimate the relationship between the California vaccine policy and changes in MMR or overall vaccine coverage, non-medical exemptions, and medical exemptions.” or similar.

8. Author summary: “What do these findings mean?”: Please revise this point to: “Our study found that the 2016 California policy to eliminate non-medical childhood vaccination exemptions was associated with increased vaccination coverage and decreased non-medical exemptions, supporting the idea that state policies can serve as effective tools to increase vaccination coverage.” or similar.

9. Introduction: Line 142: Please remove the term “significant” from the following sentence: “However, there remains significant debate around the effectiveness of policies…” as the meaning of the term is unclear.

10. Introduction: Line 144-146: Please revise to either avoid mentioning Disneyland, (e.g. edit to “Following a large measles outbreak in 2014-15, California passed…” or similar) or provide primary reference to support that the outbreak originated/was associated with Disneyland.

11. Methods: Line 286: Please reference the specific institution that waived human subjects research approval. 

12. Results: Line 375-376: Please edit sentence to “The percentage of uninsured children…” or similar.

13. Results: Line 377: Please edit “bachelor’s” to “bachelor’s degree”.

14. Results: Line 378, and where relevant throughout Methods and Results and Supporting Information: How was race/ethnicity defined and by whom? Why was race/ethnicity considered important in this study and what it is believed to represent?

15. Results: Line 381: Please remove the word “substantial” or replace with a more specific or quantitative term. 

16.Results: Line 386-387: The wording here is unclear, please revise to “Although our model found an absolute 2.4% increase in medical exemptions across all counties…” or similar.

17. Results: Lines 419-423: For the sensitivity analyses, please refer to the specific supplemental figures and tables for the discussion of the results here. 

18. Discussion: Lines 426-437: Please remove imprecise terms such as “robust” or “modest” unless these terms have specific quantitative meaning (if so please define). 

19. Discussion: Lines 434-437: Please revise this sentence to: “The use of two independent state and county level analyses with consistent results provides evidence to support the idea that adoption of similar state policies eliminating non-medical exemptions may help to address the growing public health challenge of vaccine hesitancy.” or similar.

20. Discussion: Line 466-467: Please revise to “...Mississippi and West Virginia—states that also do not allow non-medical exemptions for school…”

21. Discussion: Line 570-572: Please revise to: “Our study finds that vaccine coverage increased and non-medical exemptions decreased in the time period following the implementation of the California policy eliminating non-medical exemptions.” or similar.

22. Discussion: Line 573-574: Please revise the final sentence of the discussion to: “These study results support the idea that state-level governmental policies to remove non-medical exemptions may be effective strategies to increase vaccination coverage in the United States.” or similar.

23. Table 1: Please define the abbreviation “CI”.

24. Table 1, and throughout: Please report p values to two decimal places if p<0.05, and to three decimal places if p<0.01.

25. Table 1: Please edit “Bachelor’s” to “bachelor’s degree”.

26. Figures 1, 2, and 3: Please define the abbreviation “MMR” in the figure legend.

27. S3 Appendix: Pre-analysis Plan, Figure 1 and Table 1: Please define the abbreviation “RMSPE”.

28. S1 Table: Please provide the links to any online data sources, where feasible.

29. S1 Table: Please define abbreviations: HMO, MMR.

30. S1 Table: Please remove “bachelor’s” in “Proportion with bachelor’s high school diploma or higher”

31. S3, S4, Table: Please define the abbreviation “BS”

32. S2, S3, S4, S5 Tables and S1, S2, and S3 Figures: Please define the abbreviation “MMR”

33. S5 Table: Please explain the meaning of the row that is empty in the middle of the table.

34. S10 Table: Please define abbreviations: CI. 

35. S10 Table: “Bachelor’s” should be “bachelor’s degree”.

Comments from Reviewers:

Reviewer #1: The authors have addressed my points and clarified things.

Are they convinced that Supplementary Figure 5 does not show that changes were already occurring in California as a result of earlier legislation? That might have implications for the parallel trends assumption.

Michael Dewey

Reviewer #2: The authors have sufficiently addressed my comments.

[LINK]

---

## [Editor Report · Decision Letter 2]

20 Nov 2019

Dear Mr. Nyathi, 

On behalf of my colleagues and the academic editor, Dr. Lars Åke Persson, I am delighted to inform you that your manuscript entitled "The 2016 California Policy to Eliminate Non-Medical Vaccine Exemptions: An Empirical Policy Analysis" (PMEDICINE-D-19-02566R2) has been accepted for publication in PLOS Medicine. 

PRODUCTION PROCESS

PRESS

PROFILE INFORMATION

Thank you again for submitting the manuscript to PLOS Medicine. We look forward to publishing it. 

Best wishes, 

Caitlin Moyer, Ph.D.

Associate Editor 

PLOS Medicine

plosmedicine.org